# DocImpact: Quantifying Document Impact in RAG-LLMs

## Abstract

We present DocImpact, a novel methodology for measuring the influence of individual documents in Retrieval-Augmented Generation (RAG) systems. While RAG architectures have become increasingly popular in modern language models, understanding the precise contribution of each retrieved document to model outputs remains challenging. Our algorithm employs a counterfactual analysis by systematically excluding individual documents and measuring the divergence in model outputs compared to the full-context baseline. We implement our RAG-LLM using Pinecone as the database and Llama-3.1-70b as the LLM.

## 1 Introduction

Retrieval Augmented Generation (RAG) Lewis et al. (2020) is a natural language processing technique that enables large language models (LLMs) utilize external knowledge. The RAG process involves three components: an LLM, an external database, and a user query. The mechanism is as follows: First, we retrieve relevant data from the database based on the user query (Retrieval). Next, we augment the user query with the retrieved information to create a more comprehensive prompt (Augmentation). Finally, we feed the augmented query to our LLM to generate a result (Generation). This approach enables the LLM to produce more informed, accurate, and contextually relevant responses by leveraging external knowledge in addition to its trained knowledge.

While we can identify the specific external documents retrieved during this process, we cannot determine their influence, if any, on the final generation. It has been shown RAG-LLMs are prone to discriminatory and harmful content generation Kumar et al. (2023); Dong et al. (2024), as well as data poisoning Hong et al. (2023). Without transparency in how the retrieved documents affect the generation, we cannot properly devise a solution to prevent such problems.

The inherent complexity and scale of LLMs make it challenging to understand their decision-making process and output generation. This lack of transparency limits our ability to ensure trustworthiness, reliability, fairness, safety, and prevent hallucinations. In RAG-enabled LLMs, the retrieved documents play a more significant role in content generation than the model's training data Lewis et al. (2020). Therefore, by quantifying the influence of each retrieved document on the generated output, we can better understand the model's reasoning process and gain greater control over the qualities of the generated content. With this motivation in mind, we purpose an algorithm which quantifies the influence of each of the retrieved documents.

## 2 Related Work

Several recent works have explored and expanded the explanability of RAG systems. RAG-EX Sudhi et al. (2024) offers a model- and language-agnostic explanation framework, providing users with insights into why a large language model (LLM) might have generated a specific response. Fair-RAG Shrestha et al. (2024) addresses fairness concerns within text-to-image generative models by introducing a framework that projects reference images into the textual space. MetaRAG Zhou et al. (2024b) enhances the reasoning abilities of LLMs in multi-hop question-answering tasks by integrating retrieval-augmented generation with metacognitive strategies. For evaluating RAG systems, RAGBench Friel et al. (2024) provides a new benchmark dataset spanning various domains and

tasks, along with a novel evaluation framework called TRACe, which includes metrics like context utilization and answer completeness. Finally, a survey by the authors in Zhou et al. (2024a) explores different facets of RAGs, including transparency and fairness.

# 3 PROPOSED METHOD

The RAG-LLM process follows a systematic workflow for handling user queries. When a user submits a query, we first identify and retrieve the $k$ most relevant documents from our database. Numerous retrieval methods exist, but let us proceed with cosine similarity score due to its simplicity and widespread use. Specifically, we convert each document into a high-dimensional vector using a **word2vec** model and store both the documents and their vector representations in our database. During retrieval, the user query is similarly converted into a high-dimensional vector. We then calculate the cosine similarity between the query vector and each document vector, retrieving the top $k$ documents with the highest similarity scores. In the augmentation step, these retrieved documents are incorporated into the original user query. Finally, this augmented query is submitted to the LLM to generate a response.

Our goal is to determine how much each retrieved document has affected the LLM's response. To quantify this influence, we propose a metric called the Influence Score (IS). The IS of document $i$ ($IS_i$) is defined as follows

$$IS_i = \cos(\mathcal{F}\{G(i)\}, \mathcal{F}\{G(1, ..., k)\}) -$$
$$\cos(\mathcal{F}\{G(1, ..., i-1, i+1, ..., k)\}, \mathcal{F}\{G(1, ..., k)\}), \quad (1)$$
$$\mathcal{F} : \text{word2vec converter},$$

where the cosine measures the similarity score. Moreover, $G(1, ..., k)$, $G(1, ..., i-1, i+1, ..., k)$, and $G(i)$ denote the generated content using all $k$ documents, all $k$ documents excluding document $i$, and document $i$ only respectively. We refer to these as the original response, partial response, and individual response. We should point out that other similarity metrics such as Semantic Entropy Lin et al. (2023); Kuhn et al. (2023) could be used in place of cosine similarity as well.

To calculate the IS for all $k$ retrieved documents, we require a total of $2k + 1$ augmented generations: one generation using all $k$ documents, $k$ generations using all documents excluding one at a time, and $k$ generations using each document individually. The higher the $IS_i$, the more influence document $i$ has had in the LLM response. The need to perform $2k$ additional LLM generations introduces computational overhead, which is the drawback of our algorithm.

The rationale behind the IS definition in Equation 1 is as follows. If document $i$ has minimal influence on the **original response**, it is likely less relevant compared to other documents. In this case, its corresponding **individual response** would differ substantially from the **original response**, resulting in a small value for the first cosine term. Additionally, removing document $i$ from the augmented documents would produce a **partial response** similar to the **original response**, yielding a large value for the second cosine term. Together, these factors result in a low IS. On the other hand, if document $i$ significantly influences the **original response**, its **individual response** would closely resemble the **original response**. Furthermore, removing it from the augmented documents would yield a **partial response** that differs notably from the **original response**. These conditions lead to a high first cosine term and a low second cosine term, resulting in a high IS value.

# 4 APPLICATIONS

By having a framework that quantifies the impact of each retrieved document in the LLM response, we can pinpoint the documents responsible for each response. Specifically, it helps us with

- **Improved Fact-Checking:** By identifying the most influential documents, we can scrutinize them more closely, reducing the risk of factual errors and hallucinations in the generated response.

- **Enhanced Source Attribution:** Giving each document a clear weight helps users track where information comes from and judge how trustworthy it is.

- **Model Calibration, and Identifying Bias and Hallucination:** Analyzing document impact will help us find out what content our LLM focuses on, and as result reveal potential biases in the knowledge base and the need for calibration.

- **Document Relevance Ranking:** By quantifying document impact, we can refine retrieval algorithms, improving the quality of retrieved documents and the overall response quality.

- **Adversarial Attacks and Model Poisoning:** If our LLM produces an undesirable response, we can easily locate the responsible document and remove the poisined data.

## 5 IMPLEMENTATION

We used Pinecone as our database, `llama-3.1-70b` as our LLM, `Groq` Groq as our LLM provider, and `all-MiniLM-L6-v2` as our **word2vec** converter. The purpose of a **word2vec** converter is to map a sentence or document into numerical representations that capture their semantic and syntactic relationships, enabling metrics such as cosine to measure the similarities.

## 6 EMPIRICAL VALIDATION

To assess the functionality of our algorithm, we designed a human-in-the-loop experiment using a selected database. This experiment consists of two steps:

1. We perform a deliberate query and obtain the corresponding response, denoted **Response A**. We then rank the retrieved documents based on their IS score.

2. We repeat the same query, but this time remove the documents with the highest IS scores; denoted **Response B**.

Finally, we conduct a survey asking participants whether they perceive a significant difference between **Response A** and **Response B**. If they respond yes, then we can conclude that our algorithm has successfully identified the most influential documents.

As an empirical validation, we use a synthetic set of invoices for a retail company Kaggle. We use queries relevant to the dataset, such as "*What is the most common product bought by person X.*" For each query, we retrieved 10 documents initially. In the second step, we removed the top 3 documents with the highest IS scores.

We conducted a study with 22 participants, using a total of 12 queries. Our findings indicate that, on average, $98.86\%$ of responses showed a significant difference between the original response and the response obtained after removing the document with the highest IS.

## 7 CONCLUSION

Retrieval Augmented Generation (RAG) is a natural language processing technique that enables large language models (LLMs) utilize external knowledge. The process involves retrieving a number of documents from our database and passing them to the LLM during inference. One of the limitations of RAG is their inability to measure how individual retrieved documents affects the LLM's response. To address this, we propose Influence Score (IS), a metric that quantifies each document's impact on the LLM's output. Using our algorithm, we can pinpoint the most influential documents responsible for each response, which would help us with tasks such as fact checking and identifying biases in our LLM. The drawback of our approach is the additional computational overhead as we need to query the LLM $2k + 1$ times, where $k$ is the number of retrieved documents. We have implemented our framework, and preliminary experiments suggest that the IS highly correlates with the relevance of each document to the LLM's response.

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
