# OpenReview forum: "DocImpact: Quantifying Document Impact in RAG-LLMs"
_ICLR.cc/2025/Workshop/BuildingTrust — Submitted to BuildingTrust_

### Official Review · Reviewer_dvmW · 2025-03-01
**Paper lacks theoretical and empirical validation of the proposed method.**

**Rating:** 5
**Confidence:** 3

**Review:**

## Summary
The paper introduces an `Importance Score` to measure the impact of generation of single documents in RAG systems. The authors claim their method enables empirical identifiability of documents guiding generation.

## Strengths
- **Novelty**:  the importance score as a difference of similarity metrics between generations with and without a given document seems to be a novel idea.
- **Clarity and Organization**:  overall clear and concise explanation.
- **Impact and Relevance**:  the paper is relevant to interesting and difficult applications such as source attribution

## Weaknesses
- **Theoretical**; absence of theoretical guarantees showing the correctness or consistency of their ranking method.
- **Experimental**:  the evaluation does not provide any baseline. It seems trivial that removing 3 out of 10 documents from RAG generation changes the output of the RAG-LLM (line 142). Two baselines could consist of removing documents at random or removing the longest ones. Evaluation is restricted to one dataset/task.
- **Reproducibility**: vague implementation details regarding their evaluation pipeline. While the models and datasets are readily available, a list with all the LLM queries is missing. Only one example (line 138) is provided.
- **Computational Cost**: the computational overhead of the method seems to be quite challenging at scale, especially in the case of more complicated similarity measures such as semantic entropy which require an inner LLM sampling loop.
 - **Other**: It is not clear how to measure the impact of 2 documents with complementary information. While they can be both almost irrelevant when considered by themselves, 2 documents used together could have a higher impact than their importance score sums. Naively adapting the idea to search over the space of subsets of documents would require incredible computational costs.

## Recommendation
- **Decision**:  **Reject**
- **Key Reasons**:  weak evaluation setup and lack of theory

## Supporting Arguments
While the general idea of measuring such a similarity score seems interesting, no theory regarding a notion of correctness was provided. Moreover, the paper does not clearly show that the method performs well empirically per the lack of baselines. Removing 3 documents at random may change the generation just as much depending on the task at hand.

## Questions for the Authors
1.  How can one interpret the scale of the Importance Score? Can it be relevant in absolute terms or can it only be used for relative rankings within a prompt?
2.  For the experimental setup, why did you not use Semantic Entailment to detect changes in the RAG-LLM outputs?

## Additional Feedback
- **Suggestions for Experimental Enhancements**:  implement baselines and expand experimental setting to other datasets and RAG tasks. Use objective metrics for detecting changes in generation quality with and without the most influential documents.
- **Writing and Presentation Suggestions**: Schematics explaining the method and graphs showing its performance could be useful for visual impact and intuition.

---

### Official Review · Reviewer_m8Hw · 2025-03-02

**Rating:** 4
**Confidence:** 4

**Review:**

### Summary
This paper introduces DocImpact, a method for measuring the influence of individual documents in RAG systems. While RAG allows LLMs to integrate external knowledge during inference, the extent to which individual retrieved documents impact the final model output remains unclear. The paper proposes an IS metric that quantifies the contribution of each document by systematically excluding it from the retrieval set and measuring the divergence in the model's output.

### Strengths
- The paper is well-organized for the most part.
- The proposed metric addresses a critical gap in RAG transparency.

### Weaknesses
- The proposed method is evaluated on a synthetic invoice dataset with a small number of documents. How does the approach scale to larger document collections?
- The paper does not compare the proposed metric against existing explainability metrics in retrieval. How does IS compare to existing benchmarks for measuring retrieval impact?
- The work done so far is a promising step, but further exploration is needed to ensure completeness and robustness.

---

### Decision · Program_Chairs · 2025-03-04

Reject